# Assessing the Tumor Suppressive Impact and Regulatory Mechanisms of SPDEF Expression in Breast Cancer

**DOI:** 10.3390/cancers17213556

**Published:** 2025-11-02

**Authors:** Maansi Solanky, Maninder Khosla, Suresh K. Alahari

**Affiliations:** 1School of Medicine, Louisiana State University Health Sciences Center at New Orleans, New Orleans, LA 70112, USA; msolan@lsuhsc.edu; 2Department of Biochemistry and Molecular Biology, Louisiana State University Health Sciences Center at New Orleans, New Orleans, LA 70112, USA; 3Stanley S. Scott Cancer Center, Louisiana State University Health Sciences Center at New Orleans, New Orleans, LA 70112, USA

**Keywords:** SPDEF, breast cancer, DNA methylation, basal breast cancer, epigenetic regulation, prognostic biomarkers, tumor suppressor genes, epithelial–mesenchymal transition (EMT), precision medicine

## Abstract

**Simple Summary:**

Breast cancer is a highly diverse disease, and some tumor types, such as the Basal subtype, are particularly aggressive and insusceptible to treatment modalities. Our study investigated the SAM Pointed Domain ETS Factor (SPDEF) gene, which has been proposed to be involved in breast cell growth and differentiation. Through analysis of tumor samples from over 1200 patients, we demonstrated that low SPDEF levels were linked with worse survival across breast cancers, with the most profound changes seen in Basal tumors, younger patients, and Black or African American patients. We then determined a significant role of epigenetic DNA modification in SPDEF regulation, implicating enhanced promoter methylation underlying gene silencing. Loss of SPDEF also coincided with changes in other important pathways, including DNA repair and immune regulation. These findings suggest that SPDEF and its epigenetic regulation could serve as important clinical biomarkers and offer novel avenues for targeted therapies in breast cancer.

**Abstract:**

Background/Objectives: Breast cancer is a heterogeneous disease, and the role of the transcription factor SPDEF remains controversial. We aimed to clarify the prognostic value of SPDEF, explore demographic and molecular correlates of its expression, and investigate potential regulatory mechanisms underlying its dysregulation. Methods: Genomic and clinical data for 1218 breast cancer tumors were obtained from The Cancer Genome Atlas (TCGA). SPDEF mRNA expression was compared across intrinsic subtypes, age, and race, and prognostic significance was evaluated by Kaplan–Meier analysis. Promoter methylation patterns and DNA methyltransferase (DNMT) expression were examined as potential regulatory drivers. Co-expression analysis was performed using gene panels representing luminal differentiation, basal identity, EMT, proliferation, DNA repair, and immune signaling. Results: Low SPDEF expression was significantly associated with worse overall, relapse-free, and metastasis-free survival across all breast cancers. Expression was lowest in Basal tumors, as well as among younger and Black or African American patients. Promoter methylation at six CpG islands correlated with both reduced SPDEF expression and inferior survival, and DNMT1, DNMT3A, and DNMT3B overexpression also aligned with poor prognosis and Basal enrichment. Co-expression analysis revealed that SPDEF downregulation coincided with loss of luminal markers and increased EMT, proliferation, DNA repair, and immune pathways. Conclusions: SPDEF functions as a tumor suppressor in breast cancer, with reduced expression linked to poor outcomes, aggressive molecular features, and epigenetic regulation. These findings highlight SPDEF and DNMT-driven methylation as potential prognostic biomarkers for enhanced risk stratification and targets for novel therapies, particularly in Basal breast cancers.

## 1. Introduction

Breast cancer remains the most commonly diagnosed non-skin cancer and the second leading cause of cancer-related deaths in women in the United States [1]. In recent decades, molecular profiling, such as the PAM-50 classification system, has elucidated intrinsic tumor subtypes—Luminal A, Luminal B, HER2-enriched, Basal—each with unique prognostic and therapeutic implications [2]. Despite this enhanced understanding of pathogenesis and clinical management, heterogeneity in treatment response, acquired resistance, and survival persists [3]. In particular, Basal tumors, which largely overlap with triple-negative breast cancer (TNBC), exhibit aggressive behavior, limited therapeutic options, and poor outcomes [4]. This gap in clinical outcomes underscores the need for additional molecular biomarkers to refine prognostic assessment and identify novel therapeutic targets.

Prostate-Derived ETS Factor (SPDEF), also known as SAM Pointed Domain ETS Factor, is a transcription factor of the ETS family linked with gene regulation in epithelial and secretory tissues, such as in the breast, ovary, prostate, lung, colon, and stomach [5,6]. Notably, SPDEF expression exhibits context-dependent roles in carcinogenesis. Tumor-suppressive activity has been demonstrated in head and neck squamous cell carcinoma, colorectal cancers, and in select in vitro analyses of prostate adenocarcinoma [6,7,8,9,10]. Conversely, oncogenic activity and poor survival outcomes have also been described in prostate adenocarcinoma as well as in cancers of the pancreas, lung, and stomach [11,12,13,14]. In the realm of breast cancer, the functional significance of SPDEF remains distinctly controversial, with evidence supporting both tumor-suppressive and oncogenic functions. Moreover, subtype-specific impacts of gene expression on tumorigenesis have been reported, further complicating efforts to define SPDEF as a prognostic biomarker for breast cancer [9,14,15,16,17].

Epigenetic modifications, particularly DNA promoter methylation, represent a critical mechanism of transcriptional silencing and are frequently implicated in tumor suppressor gene inactivation [18,19,20,21]. Hypermethylation of CpG islands within promoter regions is a well-established contributor to breast cancer progression and resistance to therapy. Additionally, DNA methyltransferases (DNMTs), including DNMT1, DNMT3A, and DNMT3B, play a central role in establishing and maintaining methylation patterns, and their dysregulation is associated with aggressive tumor phenotypes and adverse outcomes [22,23].

Given these considerations, the objective of this study was threefold: (1) to assess the association between SPDEF expression and breast cancer prognosis, (2) to examine demographic and molecular correlates of SPDEF expression across intrinsic subtypes, and (3) to investigate potential regulatory mechanisms and co-expression networks linked to SPDEF expression. Through the abstraction of genomic data from The Cancer Genome Atlas Breast Cancer (TCGA BRCA) tumor registry, we show that SPDEF downregulation serves as a negative prognostic indicator in breast cancer with varying significance across molecular and demographic subtypes. In addition, we demonstrate an association between gene silencing and promoter hypermethylation, which may contribute to more aggressive clinicopathological features.

## 2. Materials and Methods

### 2.1. Data Sources

Tumor genome and patient demographic data were extracted from the TCGA BRCA registry (*n* = 1218) published on the University of California, Santa Cruz (UCSC) Xena Browser and cBio Portal [24,25,26,27]. TCGA sample IDs were utilized to match patients between the databases. Prognostic implications of SPDEF expression, methylation, and DNMT expression were assessed using Kaplan–Meier survival analysis, with differences between groups assessed by the log-rank (Mantel–Cox) test. Patients were dichotomized into high and low cohorts using the KM Plotter auto-selection methodology [28,29]. Hazard ratios (HR) with corresponding 95% confidence intervals (CI) were calculated with Cox proportional-hazards regression. Kaplan–Meier curves display survival probability with shaded 95% CIs to illustrate variability over time, and the number of patients at risk is indicated beneath each plot.

### 2.2. SPDEF mRNA Expression and Survival Analysis

SPDEF mRNA expression was reported as pan-cancer-normalized Illumina HiSeq values. Expression was analyzed across PAM-50 subtypes, age at initial pathologic diagnosis, and self-reported race categories. Cohort differences were analyzed with two-tailed Student’s *t*-test and ANOVA. For survival analysis, all breast cancer tumors were classified into high or low SPDEF expression groups determined by gene chip.

### 2.3. Regulatory Mechanism Analysis

Promoter methylation was investigated as a potential driver of SPDEF dysregulation, and data were obtained from TCGA 450K methylation arrays available on the UCSC Xena Browser. Beta values of the twelve CpG islands identified along the SPDEF promoter were evaluated. CpG islands significantly associated with SPDEF expression by Pearson correlation, as well as breast cancer survival and the Basal tumor subtype, were then isolated. A similar methodology was then employed to investigate the four DNA methyltransferase genes sequenced on the UCSC Xena Browser (https://xenabrowser.net/, accessed on 15 October 2025).

### 2.4. Differential Expression Analysis

A panel of representative genes was selected for co-expression analysis to contextualize SPDEF expression within relevant breast cancer biology. As depicted in Table 1, genes were chosen from major functional categories known to influence tumor phenotype and clinical behavior, including luminal differentiation, basal lineage, epithelial–mesenchymal transition (EMT), epithelial integrity, proliferation, DNA repair, and immune signaling. Expression differences between Basal and Non-Basal tumor subtypes were assessed using Student’s *t*-tests, and Pearson correlation analyses were performed to illuminate co-expression patterns.

Luminal transcription factors (FOXA1, GATA3, ESR1, PGR, AR, XBP1) were included as they define luminal identity and are commonly downregulated in Basal and triple-negative breast cancers [30,31]. Basal/myoepithelial genes (KRT5, KRT14, KRT17, EGFR, TP63, MIA) were chosen as markers of basal lineage [32]. EMT and stemness regulators (ZEB1, ZEB2, SNAI1, SNAI2, TWIST1, TWIST2, VIM, CDH2, CD44, ALDH1A1) were analyzed due to their established role in driving invasion, metastasis, and acquisition of stem cell–like traits in breast cancer [33,34]. Epithelial integrity genes (CDH1, CLDN3, CLDN4, CLDN7, OCLN, MUC1) were evaluated as markers of epithelial cohesion, which are frequently disrupted during EMT [35]. Proliferation regulators (MYC, CCND1, CDK4, E2F1, MKI67) were included, given the highly proliferative nature of Basal-like tumors [36]. DNA repair genes (BRCA1, BRCA2, RAD51, ATM, CHEK1, CHEK2) were evaluated because homologous recombination deficiency and BRCA-associated biology are enriched in Basal breast cancers, and prior studies identified DNA repair pathways as SPDEF-associated networks [37]. Lastly, immune checkpoint and signaling genes (CD274, CTLA4, LAG3, TIGIT, CXCL9, CXCL10, STAT1) were assessed due to the recognized immune-rich microenvironment and clinical relevance of checkpoint blockade in triple-negative/Basal breast cancers [38,39].

Incorporation of these functional categories enabled investigation of molecular correlates of SPDEF downregulation. We aimed to understand if SPDEF expression patterns in Basal tumors coincided with loss of luminal differentiation and gain of EMT, proliferative, DNA repair, and immune activation programs.

### 2.5. Statistical Analysis

All statistical analyses were performed using GraphPad Prism (v9) and Microsoft Excel. A *p*-value < 0.05 was considered statistically significant.

## 3. Results

### 3.1. Prognostic Significance of SPDEF Expression in Breast Cancer Tumors

Kaplan–Meier analyses for overall survival (OS), relapse-free survival (RFS), distant metastasis-free survival (DMFS), and palliative performance scale (PPS) were conducted to evaluate the relationship between SPDEF expression and clinicopathological outcomes. Results are depicted in Figure 1. Patients with higher SPDEF expression consistently exhibited superior outcomes compared to those with low expression (OS: HR = 0.68, RFS: HR = 0.68, DMFS: HR = 0.69, PPS: HR = 0.68; all *p* < 0.01). These findings indicate a strong directional association wherein loss of SPDEF expression correlates with poorer short and long-term prognosis. Of note, RFS, DMFS, and PPS demonstrated consistently inferior outcomes in patients with low SPDEF expression through the analyzed timeframe, suggesting short- and long-term impacts on morbidity. In contrast, a stark difference in overall survival emerged around 80 months of follow-up, indicating a more long-term mortality association.

### 3.2. Demographic and Molecular Correlates of SPDEF Expression

To determine trends in SPDEF variation, gene expression was then compared between cohorts of tumor subtype, race category, and age at initial pathologic diagnosis. Stratification by intrinsic subtype revealed that average SPDEF expression was markedly lower in Basal tumors compared with Luminal A, Luminal B, HER2-enriched, and Normal subtypes (*p* < 0.0001) (Figure 2). Interestingly, other subtypes—Luminal A, Luminal B, Her2-enriched—exhibited higher average SPDEF expressions than the Normal subtype. Demographic comparisons also yielded important expression trends. Average SPDEF levels were significantly lower in tumors of Black and African American patients and those of younger diagnosis ages (Figure 3). These associations suggest that SPDEF silencing disproportionately affects clinically aggressive tumor subtypes and certain demographic groups. All demographic analyses, along with receptor statuses are summarized in Table 2.

### 3.3. Promoter Methylation

The twelve CpG islands recognized along the SPDEF promoter were examined to gain insight into the mechanism underlying gene expression. Pearson correlation analysis determined a significant linear relationship between methylation and SPDEF expression among ten loci; results are summarized in Table 3. Kaplan–Meier analysis was then performed to examine clinical significance, and increased methylation at seven CpG islands corresponded with notably greater survival hazard (Figure 4). A majority of CpG islands exhibited higher methylation levels amongst Basal tumors compared with Non-Basal tumors, aligning with the subtype’s previously observed expression pattern (Table 4). These findings support promoter hypermethylation as a mechanism of SPDEF downregulation in breast cancers.

### 3.4. DNMT Expression

A similar analysis was performed to evaluate the role of DNA methyltransferases. Expression of all DNMT genes was found to significantly correlate with SPDEF expression and demonstrate strong inverse linear relationships (DNMT1: r = −0.13, DNMT3A: r = −0.10, DNMT3B: r = −0.12, DNMT3L: r = −0.17; all *p* < 0.001), as depicted in Figure 5. However, interpretation of DNMT3L expression is constrained by suspected data error, with multiple entries showing an identical value of −0.505. Elevated expression of DNMTs, excluding DNMT3L, was also found to significantly correspond with adverse survival outcomes, and Kaplan–Meier analyses are included in Figure 6. Additionally, expression of all DNMTs was uniformly highest among Basal tumors (Table 5), further reinforcing the presence of a methylation-driven regulatory model.

### 3.5. Differential Expression Analysis

A curated panel enabled investigation into whether SPDEF downregulation in Basal tumors coincides with loss of luminal differentiation and gain of EMT, proliferative, DNA repair, and immune activation programs. Linear regression models were first fit between the expression of SPDEF and each of the selected genes. This offered insight into the significance and direction of tumor co-expression patterns, as demonstrated in Table 6. Then, the average expression of each gene within Basal and Non-Basal tumor subtypes was investigated; results from this cohort analysis are presented in Table 7.

Luminal-defining transcription factors (FOXA1, GATA3, ESR1, PGR, AR, XBP1) were all found to be uniformly reduced in Basal compared with Non-Basal tumors (all *p* < 0.001). Similarly, two epithelial markers (OCLN, MUC1) were also decreased in Basal tumors (both *p* < 0.001). These findings highlight that SPDEF silencing is paralleled by a loss of luminal identity, indicating the possible role of SPDEF as a marker of epithelial differentiation. All Basal/myoepithelial markers, excluding TP63, were significantly elevated in Basal tumors (all *p* < 0.001), consistent with current understanding. Select EMT and stemness regulators (SNAI1, VIM, CDH2, CD44) were also increased in Basal tumors (all *p* < 0.05). Together, these relationships suggest that loss of SPDEF expression is accompanied by acquisition of mesenchymal and stem-like characteristics.

Cell cycle and proliferation markers (MYC, CCND1, CDK4, E2F1, MKI67) as well as DNA regulators (BRCA2, RAD51, CHEK1, CHEK2) were significantly higher in Basal compared to Non-Basal tumors (all *p* < 0.001). This pattern underscores the highly proliferative nature of Basal tumors and suggests a possible link between SPDEF loss and cell cycle deregulation. Furthermore, immune-related genes (CD274, CTLA4, LAG3, TIGIT, CXCL9, CXCL10, STAT1) were also elevated amongst tumors of the Basal subtype (all *p* < 0.01).

Expression of these genes was then plotted alongside SPDEF expression within Basal and Non-Basal tumor subtypes to identify the presence of a directional relationship; results of linear correlation analysis are shown below in Table 8. Within the Basal subtype, direct linear relationships were found between expression of SPDEF and expression of FOXA1, PGR, AR, and XBP1. Indirect linear relationships with SPDEF were found with TWIST2, CDH2, MYC, CCND1, MKI67, BRCA2, RAD51, CD274, LAG3, CXCL10, and STAT1.

## 4. Discussion

SPDEF (SAM Pointed Domain ETS Transcription Factor) is a transcription factor normally expressed in epithelial tissues, where it contributes to cellular differentiation, epithelial polarity, and maintenance of glandular architecture. In breast epithelium, it functions downstream of luminal lineage transcription factors and promotes terminal differentiation of ER-positive cells. This role has been implicated with invasive breast cancer in opposing ways, dependent on tumor subtype. In luminal cells, loss of SPDEF has been associated with decreased luminal differentiation and subsequently increased tumor survival and endocrine therapy resistance. Meanwhile, in Basal cells, loss of SPDEF has been associated with increased tumor survival and invasiveness [5,6,15,17,37].

This study provides compelling evidence that SPDEF functions as a tumor suppressor in breast cancer, particularly among tumors of Basal cells, with low expression strongly associated with adverse clinical and molecular outcomes. Our analysis revealed a significant directional association between low SPDEF expression and reduced survival, providing more evidence to supplement previous work examining the transcription factor’s controversial role. We contribute to these studies by isolating possible pathological mechanisms that underlie these findings.

Our results are consistent with previous studies reporting tumor-suppressive properties of SPDEF in cancers of the prostate, colon, and head and neck, where its loss was found to contribute to more aggressive disease phenotypes [6,7,8,9,10]. In contrast, other reports have suggested oncogenic activity of SPDEF [11,12,13,14], underscoring the context-dependent nature of SPDEF’s function across and within cancer types. Within the realm of breast cancer, our results align more closely with those observing anti-oncogenic effects of SPDEF, suggesting that SPDEF loss in breast tissue promotes progression, particularly in aggressive tumor subtypes.

Importantly, we identified striking molecular and demographic patterns associated with SPDEF silencing. Low SPDEF expression was most pronounced in Basal tumors, a subtype broadly characterized by poor prognosis and limited therapeutic options. Furthermore, decreased expression was more prevalent among younger patients and those identifying as Black or African American. These observations align with documented disparities in breast cancer outcomes and may indicate a biological component contributing to differences in prognosis [38,39,40].

Mechanistically, our findings strongly implicate promoter methylation as a key driver of SPDEF silencing. A summary of our methylation analyses and framework for interpretation is depicted in Figure 7. Methylation at seven of the twelve CpG islands along the SPDEF promoter was found to inversely correlate with SPDEF expression. Six of these CpG islands additionally exhibited correlation with inferior breast cancer survival as well as elevated methylation in Basal tumors. Increased methylation among these Basal tumors suggests an epigenetically mediated mechanism contributing to aggressive disease biology. Additionally, the concurrent overexpression of DNMT1, DNMT3A, DNMT3B, and DNMT3L in low-SPDEF tumors reinforces the role of DNA methyltransferases in maintaining hypermethylation states. This is consistent with prior evidence that DNMT-mediated methylation orchestrates silencing of tumor suppressor genes and promotes oncogenesis [21,22,23].

In addition, our analysis of differential gene expression between Basal and Non-Basal tumor subtypes provides considerable evidence that SPDEF downregulation is a defining molecular feature of the Basal subtype. Basal tumors displayed markedly lower SPDEF expression compared to Non-Basal tumors, accompanied by broad loss of luminal differentiation markers and upregulation of Basal lineage, EMT/stemness, proliferative, DNA repair, and immune-related pathways. These results position SPDEF as a possible molecular gatekeeper of luminal epithelial identity, with its loss facilitating dedifferentiation, EMT, and activation of pathways linked to poor prognosis.

Our findings have important translational implications. SPDEF expression and methylation status may serve as prognostic biomarkers across breast cancers, reflecting adverse clinical outcomes and aggressive molecular behavior. The observed DNMT overexpression in these patients highlights the potential for epigenetic therapies, such as DNMT inhibitors, to restore SPDEF expression and improve disease trajectory. In addition, co-expression trends identified with SPDEF downregulation suggest promise in novel therapeutics that target enrichment of DNA repair and immune-related markers. Notably, these relationships appear most pronounced within the Basal subtype, where SPDEF downregulation is more frequent, indicating that this group may be particularly susceptible to such interventions.

In addition, recognition of promoter methylation as a key regulatory mechanism offers practical clinical utility. The recent literature regarding prostate cancer has demonstrated that SPDEF CpG island methylation can be reliably detected in peripheral blood through PCR-based assays [41], enabling a non-invasive and tumor-specific approach to supplement diagnostic and therapeutic management. A comparable liquid biopsy approach could be applied in breast cancer to enhance risk stratification, monitor disease progression, and guide the use of epigenetic therapies. Ultimately, integration of *SPDEF*-related biomarkers into precision medicine frameworks may inform treatment strategies tailored to both overall breast cancer prognosis and subtype-specific risk profiles.

## 5. Limitations

This study is based on retrospective analysis of publicly available datasets and thereby lacks direct experimental validation and may introduce selection and reporting biases. Additionally, functional validation of SPDEF’s tumor-suppressive role was beyond the scope of this study. Future research should focus on experimental confirmation of these findings and investigation of therapeutic strategies targeting SPDEF methylation.

## 6. Conclusions

Collectively, our results identify SPDEF as a tumor suppressor gene in breast cancer, with decreased expression significantly associated with poor survival, aggressive tumor subtypes, and specific demographic patterns. Promoter hypermethylation, mediated by DNMT enzymes, emerges as a possible mechanism regulating SPDEF expression. Moreover, SPDEF downregulation is associated with loss of luminal lineage, EMT, and activation of DNA repair and immune protection. These insights underscore the prognostic and therapeutic relevance of SPDEF and its epigenetic modifiers, paving the way for novel biomarker-driven strategies and epigenetic interventions in breast cancer management.

## Figures and Tables

**Figure 1 cancers-17-03556-f001:**
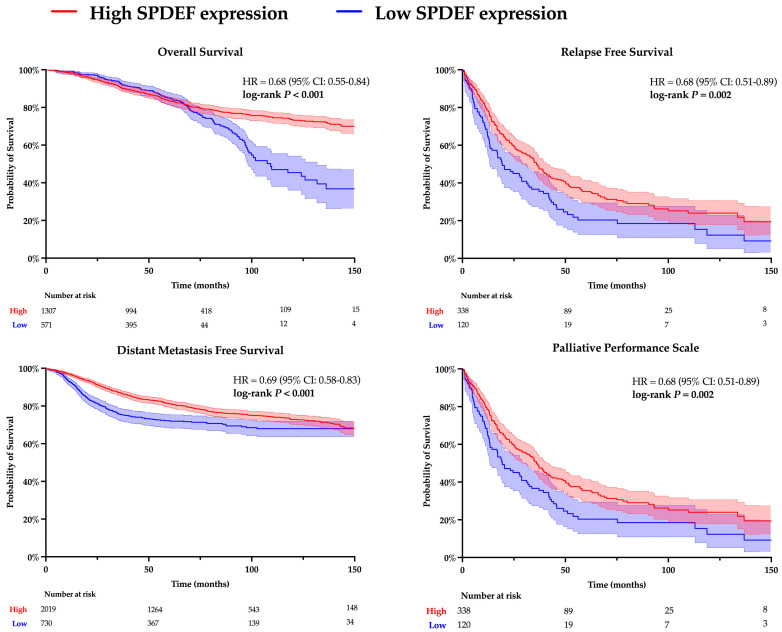
Kaplan–Meier analysis comparing high and low SPDEF expression groups. Patients were dichotomized into higher SPDEF expression (red) and lower SPDEF expression (blue) groups. Curves represent estimated probability over time with shaded 95% confidence intervals for overall survival, relapse-free survival, distant metastasis-free survival, and palliative performance scale. Hazard ratios (HR) with 95% confidence intervals and log-rank *p*-values are displayed within each panel. The number of patients remaining at risk is shown below the *X*-axis.

**Figure 2 cancers-17-03556-f002:**
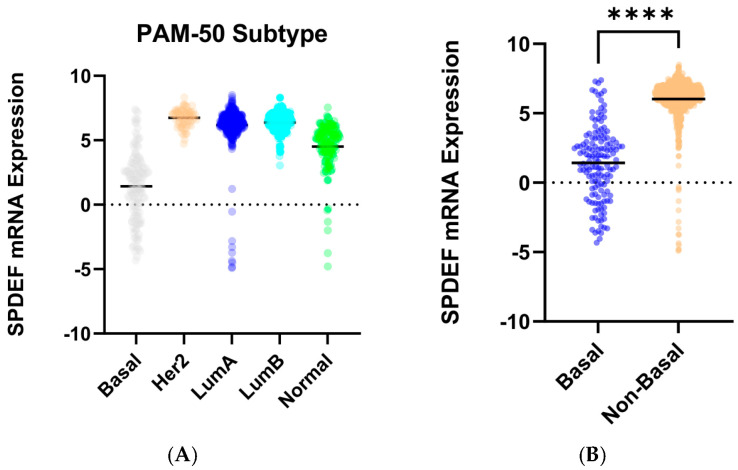
SPDEF mRNA expression across intrinsic breast cancer subtypes. (**A**) displays SPDEF expression values stratified by PAM-50 subtype (Basal, Her2-enriched, Luminal A, Luminal B, and Normal). Each dot represents an individual tumor, and horizontal black bars denote mean expression within each subtype. (**B**) compares Basal tumors to a pooled Non-Basal group (Her2-enriched, Luminal A, Luminal B, and Normal). Basal tumors demonstrate significantly reduced SPDEF expression compared to Non-Basal tumors (*p* < 0.001), which is indicated with asterisks.

**Figure 3 cancers-17-03556-f003:**
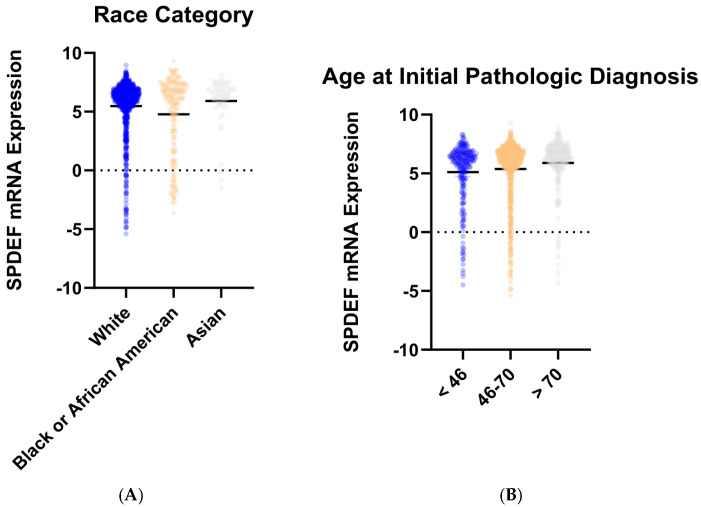
SPDEF mRNA expression stratified by demographic features. (**A**) depicts SPDEF expression across self-reported race categories (White, Black or African American, Asian). (**B**) depicts SPDEF expression amongst three categories of age at initial pathologic diagnosis (<46 years, 46–70 years, >70 years). Each dot represents a patient, and horizontal black bars denote mean SPDEF expression within each demographic category.

**Figure 4 cancers-17-03556-f004:**
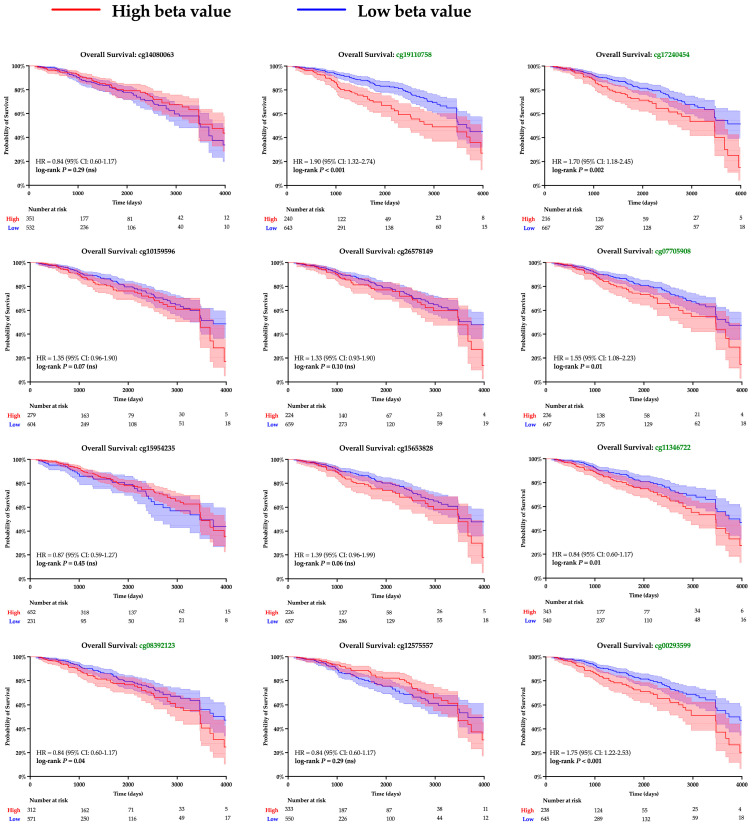
Kaplan–Meier survival analysis of SPDEF promoter methylation at CpG islands. Kaplan–Meier survival curves are shown for patients with tumors stratified into higher (red) and lower (blue) levels of methylation at each CpG island within the SPDEF promoter. Methylation was quantified with beta values, and loci with significant log-rank *p*-values are demarked with green font. 95% confidence intervals are shaded around both curves, and the number of patients remaining at risk is included beneath each respective *X*-axis.

**Figure 5 cancers-17-03556-f005:**
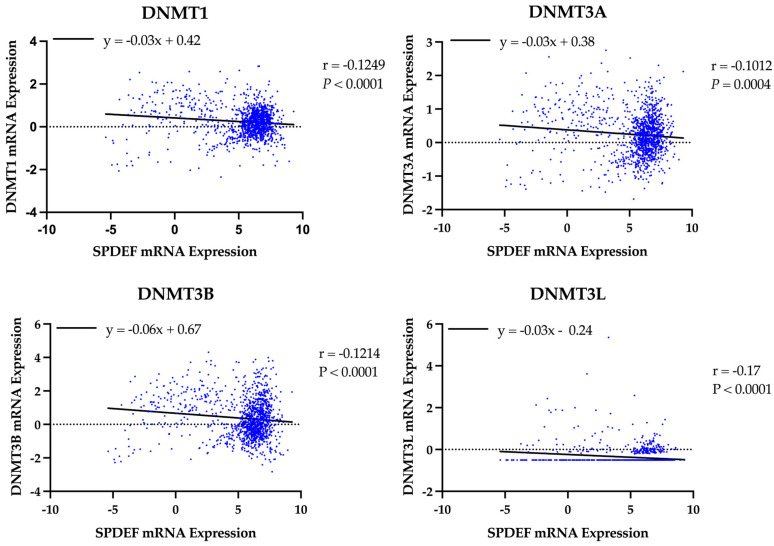
Correlation of SPDEF expression with DNA methyltransferase expression. Scatterplots display the relationship between SPDEF expression and DNMT1, DNMT3A, DNMT3B, and DNMT3L across all tumor samples. Lines of best-fit are included. Pearson correlation coefficients (r) and *p*-values are indicated.

**Figure 6 cancers-17-03556-f006:**
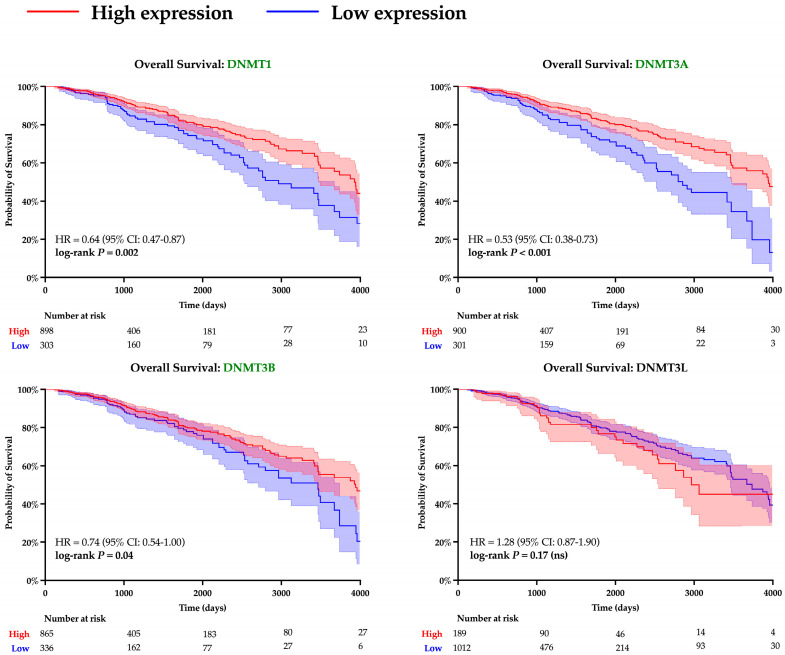
Kaplan–Meier survival analysis of DNMT expression. Survival curves are shown for patients with tumors stratified into high (red) and low (blue) levels of expression of various DNMTs. Statistical differences in hazard were computed with log-rank testing, and curves with significant *p*-values are denoted with green font. Shading around each curve represents 95% confidence intervals, and number of patients remaining at risk is depicted for each cohort beneath *X*-axes.

**Figure 7 cancers-17-03556-f007:**
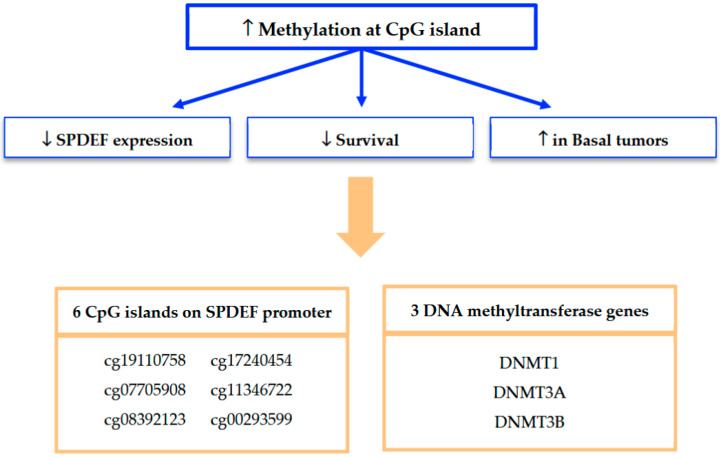
Summary of findings from SPDEF methylation analysis. Methylation at seven of twelve CpG islands was found to significantly associate with reduced SPDEF expression and inferior survival outcomes; six of these regulatory loci were also associated with increased methylation among Basal tumors. Expression of three DNMT genes was also found to significantly associate with reduced SPDEF expression, inferior survival outcomes, and the Basal tumor subtype.

**Table 1 cancers-17-03556-t001:** Representative gene panel selected for differential expression analysis. Genes were grouped into seven categories to capture various biological hallmarks of breast cancer, including luminal differentiation, basal identity, EMT, proliferative activity, DNA repair capacity, and immune microenvironment signaling.

Gene Panel	
Luminal transcription factors	FOXA1, GATA3, ESR1, PGR, AR, XBP1
Basal/myoepithelial differentiation markers	KRT5, KRT14, KRT17, EGFR, TP63, MIA
EMT/stemness regulators	ZEB1, ZEB2, SNAI1, SNAI2, TWIST1, TWIST2, VIM, CDH2, CD44, ALDH1A1
Epithelial integrity markers	CDH1, CLDN3, CLDN4, CLDN7, MUC1
Proliferation genes	MYC, CCND1, CDK4, E2F1, MKI67
DNA repair genes	BRCA1, BRCA2, RAD51, ATM, CHEK1, CHEK2
Immune checkpoint and signaling genes	CD274, CTLA4, LAG3, TIGIT, CXCL9, CXCL10, STAT1

**Table 2 cancers-17-03556-t002:** Average SPDEF expression across molecular and demographic subgroups. SPDEF mRNA expression values are summarized by PAM-50 subtype, age at initial pathologic diagnosis, race category, and receptor status (ER, PR, HER2). *p*-values represent the overall comparison of mean SPDEF expression across subgroups within each category, generated by one-way ANOVA (for multiple group comparisons) or Student’s *t*-test (for two group comparisons). Significant *p*-values are bolded.

	*n*	Average SPDEF Expression	*p*-Value
**PAM-50 subtype**			**<0.001**
Basal-like	142	1.427	
Her2-enriched	67	6.740	
Luminal A	434	6.178	
Luminal B	194	6.370	
Normal-like	119	4.515	
**Age at initial pathologic diagnosis**			**0.003**
<46 years	222	5.107	
46–70 years	761	5.372	
>70 years	232	5.891	
**Race category**			**0.001**
White	753	5.472	
Black or African American	177	4.773	
Asian	59	5.910	
**ER status**			**<0.001**
ER Negative	260	3.035	
ER Positive	892	6.151	
**PR status**			**<0.001**
PR Negative	376	3.988	
PR Positive	773	6.167	
**HER2 status**			**<0.001**
HER2 Negative	652	5.345	
HER2 Positive	114	6.486	

**Table 3 cancers-17-03556-t003:** Correlations between CpG site methylation and SPDEF expression. Pearson correlation coefficients (r) and *p*-values are shown for each of 12 CpG islands determined in the SPDEF promoter. Significant negative correlations (*p* < 0.05) indicate that greater methylation in that site was associated with reduced SPDEF expression. Significance is demarked with bold font.

	r	*p*-Value
cg14080063	−0.047	0.169
cg19110758	−0.377	**<0.001**
cg17240454	−0.654	**<0.001**
cg10159596	−0.246	**<0.001**
cg26578149	−0.235	**<0.001**
cg07705908	−0.547	**<0.001**
cg15954235	−0.305	**<0.001**
cg15653828	−0.537	**<0.001**
cg11346722	−0.606	**<0.001**
cg08392123	−0.630	**<0.001**
cg12575557	−0.025	0.459
cg00293599	−0.657	**<0.001**

**Table 4 cancers-17-03556-t004:** Average methylation of SPDEF promoter CpG islands for Basal and Non-Basal tumors. Methylation at each site was quantified with beta values. Student’s *t*-test was performed to establish significant differences between tumor types, and *p*-values are reported. Significant *p*-values are bolded.

	Basal	Non-Basal	*p*-Value
cg14080063	0.898	0.896	0.751
cg19110758	0.596	0.487	**<0.001**
cg17240454	0.646	0.474	**<0.001**
cg10159596	0.290	0.282	0.404
cg26578149	0.309	0.308	0.847
cg07705908	0.467	0.388	**<0.001**
cg15954235	0.194	0.179	0.139
cg15653828	0.487	0.387	**<0.001**
cg11346722	0.906	0.568	**<0.001**
cg08392123	0.937	0.690	**<0.001**
cg12575557	0.945	0.946	0.635
cg00293599	0.642	0.446	**<0.001**

**Table 5 cancers-17-03556-t005:** Average expression of DNMT genes among Basal and Non-Basal tumors. Student’s *t*-test was performed to quantify differences between tumor types, and *p*-values are reported. Significant *p*-values are bolded.

	Basal	Non-Basal	*p*-Value
DNMT1	0.875	0.115	**<0.001**
DNMT3A	0.849	0.109	**<0.001**
DNMT3B	1.547	0.155	**<0.001**
DNMT3L	−0.212	−0.428	**<0.001**

**Table 6 cancers-17-03556-t006:** Correlation of SPDEF expression and expression of representative genes. Pearson correlation coefficients (r) and *p*-values are reported for each gene’s association with SPDEF expression. Positive correlations indicate that higher SPDEF expression is associated with higher expression of the corresponding gene, whereas negative correlations indicate inverse relationships. Direction and significance of linear relationships are summarized in the “Relationship” column. Gene families are distinguished with shading, and significant *p*-values are bolded.

Gene	r	*p*-Value	Relationship	Gene	r	*p*-Value	Relationship
**FOXA1**	0.871	**<0.001**	**+**	**PGR**	0.441	**<0.001**	**+**
**GATA3**	0.718	**<0.001**	**+**	**AR**	0.544	**<0.001**	**+**
**ESR1**	0.600	**<0.001**	**+**	**XBP1**	0.776	**<0.001**	**+**
**KRT5**	−0.263	**<0.001**	**−**	**EGFR**	−0.486	**<0.001**	**−**
**KRT14**	−0.169	**<0.001**	**−**	**TP63**	0.041	0.149	ns
**KRT17**	−0.205	**<0.001**	**−**	**MIA**	−0.393	**<0.001**	**−**
**ZEB1**	0.059	**0.041**	**+**	**TWIST2**	−0.182	**<0.001**	**−**
**ZEB2**	−0.230	**<0.001**	**−**	**VIM**	−0.377	**<0.001**	**−**
**SNAI1**	−0.373	**<0.001**	**−**	**CDH2**	−0.064	**0.026**	**−**
**SNAI2**	−0.156	**<0.001**	**−**	**CD44**	−0.100	**0.001**	**−**
**TWIST1**	−0.118	**<0.001**	**−**	**ALDH1A1**	−0.072	**0.012**	**−**
**CDH1**	0.259	**<0.001**	**+**	**CLDN7**	0.378	**<0.001**	**+**
**CLDN3**	0.375	**<0.001**	**+**	**OCLN**	0.306	**<0.001**	**+**
**CLDN4**	0.252	**<0.001**	**+**	**MUC1**	0.481	**<0.001**	**+**
**MYC**	−0.375	**<0.001**	**−**	**E2F1**	−0.078	**0.007**	**−**
**CCND1**	0.336	**<0.001**	**+**	**MKI67**	−0.114	**<0.001**	**−**
**CDK4**	−0.094	**0.001**	**−**				
**BRCA1**	0.087	**0.003**	**+**	**ATM**	−0.142	**<0.001**	**−**
**BRCA2**	−0.132	**<0.001**	**−**	**CHEK1**	−0.290	**<0.001**	**−**
**RAD51**	−0.076	**0.008**	**−**	**CHEK2**	−0.201	**<0.001**	**−**
**CD274**	−0.223	**<0.001**	**−**	**CXCL9**	−0.096	**0.008**	**−**
**CTLA4**	−0.200	**<0.001**	**−**	**CXCL10**	−0.187	**<0.001**	**−**
**LAG3**	−0.201	**<0.001**	**−**	**STAT1**	−0.152	**<0.001**	**−**
**TIGIT**	−0.137	**<0.001**	**−**				

**Table 7 cancers-17-03556-t007:** Average expression of representative genes in Basal vs. Non-Basal tumors. Average pancancer-normalized mRNA expression values for the above genes are presented for Basal and Non-Basal tumors. Differences between Basal and Non-Basal tumors were examined with the two-tailed Student’s *t*-test, and *p*-values have been included. Significant differences in gene expression indicate widespread molecular reprogramming in Basal tumors. Gene families are distinguished with shading, and significant *p*-values are bolded.

Average mRNA Expression							
Gene	Basal	Non-Basal	*p*-Value	Gene	Basal	Non-Basal	*p*-Value
**FOXA1**	0.121	5.880	**<0.001**	**PGR**	−0.483	4.914	**<0.001**
**GATA3**	2.655	6.482	**<0.001**	**AR**	−0.636	4.475	**<0.001**
**ESR1**	0.100	6.310	**<0.001**	**XBP1**	0.155	3.019	**<0.001**
**KRT5**	4.562	1.967	**<0.001**	**EGFR**	0.280	−0.179	**<0.001**
**KRT14**	4.642	3.121	**<0.001**	**TP63**	0.083	1.235	**<0.001**
**KRT17**	3.045	0.384	**<0.001**	**MIA**	4.850	0.726	**<0.001**
**ZEB1**	−0.761	0.453	**<0.001**	**TWIST2**	0.360	0.725	**0.004**
**ZEB2**	−0.423	0.026	**<0.001**	**VIM**	0.493	0.085	**<0.001**
**SNAI1**	1.242	0.303	**<0.001**	**CDH2**	−0.559	−1.213	**0.001**
**SNAI2**	0.745	0.824	0.462	**CD44**	0.866	0.489	**<0.001**
**TWIST1**	0.932	1.238	**0.011**	**ALDH1A1**	−2.305	−1.359	**<0.001**
**CDH1**	1.482	1.619	0.229	**CLDN7**	1.170	1.250	0.540
**CLDN3**	2.002	1.893	0.541	**OCLN**	0.663	1.221	**<0.001**
**CLDN4**	2.151	1.438	**<0.001**	**MUC1**	0.857	2.463	**<0.001**
**MYC**	0.772	−0.125	**<0.001**	**E2F1**	1.000	−0.477	**<0.001**
**CCND1**	−0.479	1.174	**<0.001**	**MKI67**	2.034	0.249	**<0.001**
**CDK4**	0.192	−0.299	**<0.001**				
**BRCA1**	0.110	0.334	0.091	**ATM**	−0.011	0.175	**0.006**
**BRCA2**	1.547	0.695	**<0.001**	**CHEK1**	1.214	−0.325	**<0.001**
**RAD51**	1.482	0.080	**<0.001**	**CHEK2**	1.101	−0.080	**<0.001**
**CD274**	0.080	−0.353	**0.004**	**CXCL9**	2.326	1.044	**<0.001**
**CTLA4**	1.071	−0.241	**<0.001**	**CXCL10**	2.702	0.691	**<0.001**
**LAG3**	0.799	−0.708	**<0.001**	**STAT1**	0.818	0.343	**<0.001**
**TIGIT**	1.432	0.185	**<0.001**				

**Table 8 cancers-17-03556-t008:** Correlation of SPDEF expression and expression of representative genes within Basal vs. Non-Basal tumors. Pearson correlation coefficients (r) and *p*-values are reported for each gene’s association with SPDEF expression, analyzed separately in Basal and Non-Basal tumors. Positive correlations indicate that higher SPDEF expression is associated with higher expression of the corresponding gene; whereas, negative correlations indicate inverse relationships. Gene families are distinguished with shading, and significant *p*-values are bolded.

Gene	Basal		Non-Basal		Gene	Basal		Non-Basal	
	r	*p*-Value	R	*p*-Value		r	*p*-Value	R	*p*-Value
**FOXA1**	0.512	**<0.001**	0.867	**<0.001**	**PGR**	0.247	**0.003**	0.016	0.658
**GATA3**	0.104	0.220	0.610	**<0.001**	**AR**	0.205	**0.014**	0.060	0.088
**ESR1**	0.026	0.764	0.264	**<0.001**	**XBP1**	0.392	**<0.001**	0.647	**<0.001**
**KRT5**	−0.129	0.126	−0.109	**0.002**	**EGFR**	−0.162	0.054	−0.444	**<0.001**
**KRT14**	−0.123	0.144	−0.093	**0.008**	**TP63**	0.107	0.207	−0.140	**<0.001**
**KRT17**	−0.038	0.653	−0.007	0.844	**MIA**	−0.042	0.620	−0.135	**<0.001**
**ZEB1**	0.025	0.767	−0.441	**<0.001**	**TWIST2**	−0.229	**0.006**	−0.448	**<0.001**
**ZEB2**	−0.144	0.087	−0.576	**<0.001**	**VIM**	−0.026	0.759	−0.497	**<0.001**
**SNAI1**	−0.081	0.340	−0.161	**<0.001**	**CDH2**	−0.171	**0.043**	0.194	**<0.001**
**SNAI2**	0.018	0.833	−0.220	**<0.001**	**CD44**	0.130	0.122	−0.041	0.238
**TWIST1**	0.007	0.938	−0.315	**<0.001**	**ALDH1A1**	0.012	0.889	−0.482	**<0.001**
**CDH1**	0.091	0.281	0.523	**<0.001**	**CLDN7**	0.058	0.494	0.769	**<0.001**
**CLDN3**	0.127	0.131	0.675	**<0.001**	**OCLN**	−0.135	0.110	0.492	**<0.001**
**CLDN4**	0.148	0.080	0.677	**<0.001**	**MUC1**	0.048	0.571	0.546	**<0.001**
**MYC**	−0.307	**<0.001**	−0.248	**<0.001**	**E2F1**	−0.040	0.641	0.402	**<0.001**
**CCND1**	−0.230	**0.006**	0.197	**<0.001**	**MKI67**	−0.307	**0.002**	0.428	**<0.001**
**CDK4**	0.117	0.166	0.294	**<0.001**					
**BRCA1**	−0.020	0.812	0.169	**<0.001**	**ATM**	−0.099	0.240	−0.353	**<0.001**
**BRCA2**	−0.268	**0.001**	0.247	**<0.001**	**CHEK1**	−0.049	0.561	0.291	**<0.001**
**RAD51**	−0.180	**0.032**	0.444	**<0.001**	**CHEK2**	0.021	0.806	0.333	**<0.001**
**CD274**	−0.259	**0.002**	−0.175	**<0.001**	**CXCL9**	−0.060	0.477	0.142	**<0.001**
**CTLA4**	−0.165	0.050	0.038	0.277	**CXCL10**	−0.245	**0.003**	0.237	**<0.001**
**LAG3**	−0.203	**0.015**	0.116	**0.001**	**STAT1**	−0.327	**<0.001**	0.090	**0.010**
**TIGIT**	−0.097	0.253	0.105	**0.003**					

## Data Availability

cBio Portal for Cancer Genomics: https://www.cbioportal.org/ (accessed on 15 October 2025); UCSC Xena Functional Genomics Explorer: https://xenabrowser.net/ (accessed on 15 October 2025); Kaplan–Meier Plotter: https://kmplot.com/analysis/ (accessed on 15 October 2025).

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
