# Peer review of "Assessing the Tumor Suppressive Impact and Regulatory Mechanisms of SPDEF Expression in Breast Cancer"

_cancers, 2025, doi:10.3390/cancers17213556_

Round 1
Reviewer 1 Report
Comments and Suggestions for Authors
1. The information in Table 1 and the text following the table is completely duplicated. Either add the appropriate references to the table or delete the text following the table.
2. The Kaplan-Meier curves (Figs. 1, 4, and 6) should be plotted with a range of variation to understand the significance of the differences. Please add this. Despite the statistical significance of the differences, high SPDEF expression improves survival rates over short follow-up periods; after 150 months, the differences are not obvious.
3. Table 2 - is it really possible to evaluate SPDEF expression with such a large number of decimal places? Are the p-values ​​used to compare which subgroups? Please clarify this in the caption below the table.
4. In Table 6, is the BRCA1 row a typo – the value is 3340? Many numbers are presented with varying precision; please round them appropriately. A similar comment is also made in Table 7.
5. Figure 7 shows 6 CpG islands, while 7 are listed.
6. I find the conclusion about therapeutic prospects questionable, since the finding that reduced SPDEF expression in basal-like cancer correlates with a poor prognosis adds nothing to clinical practice. Describe the prospects in more detail, using specific examples rather than generalities.
Author Response
- The information in Table 1 and the text following the table is completely duplicated. Either add the appropriate references to the table or delete the text following the table. Response: We added this table as a concise visual to depict which genes are associated with each category. I can remove it if it is redundant.
- The Kaplan-Meier curves (Figs. 1, 4, and 6) should be plotted with a range of variation to understand the significance of the differences. Please add this. Despite the statistical significance of the differences, high SPDEF expression improves survival rates over short follow-up periods; after 150 months, the differences are not obvious. Response: Good Suggestions. All Kaplan-Meier curves have been updated with shading to depict 95% confidence intervals, and timeframes have been adjusted to better capture each dataset.
3. Table 2 - is it really possible to evaluate SPDEF expression with such a large number of decimal places? Are the p-values ​​used to compare which subgroups? Please clarify this in the caption below the table. Response: This is a great suggestion, and thus I changed numbers to have 3 decimal places. I have added a clarification regarding the P-values to the Table 2 caption.
4. In Table 6, is the BRCA1 row a typo – the value is 3340? Many numbers are presented with varying precision; please round them appropriately. A similar comment is also made in Table 7.Response: Yes, it's a typo, and we apologize for this and has been fixed. As mentioned in #3, we have standardized precision to 3 decimal places.
5. Figure 7 shows 6 CpG islands, while 7 are listed. Response: WE apologize for this typo and has been corrected. There should only be 6.
6. I find the conclusion about therapeutic prospects questionable, since the finding that reduced SPDEF expression in basal-like cancer correlates with a poor prognosis adds nothing to clinical practice. Describe the prospects in more detail, using specific examples rather than generalities. Response: Great Suggestions. We have addressed this in the discussion section and added one extra paragraph.
Reviewer 2 Report
Comments and Suggestions for Authors
Study „ Assessing the Tumor Suppressive Impact and Regulatory Mechanisms of SPDEF Expression in Breast Cancer“ by Solanky et al. evaluates the expression of SPDEF transcription factor in breast cancer and its relation with patient survival and expression of genes involved in EMT, DNA replication and repair and immune regulation. The authors associate its reduced expression with poor outcomes, aggressive molecular features, and epigenetic regulation. Epigenetic silencing of SPDEF was in this study associated with overexpression of DNA methyltransferases Co-expression analysis revealed that SPDEF down-regulation coincided with loss of luminal markers and increased EMT, proliferation, DNA repair, and immune pathways.
The study is very good conceptualized and the manuscript is well written.
However, I have few suggestions.
I think that BRCA is not appropriate abbreviation for breast cancer as it is usually related to BRCA1 and BRCA2 genes.
Also, in Discussion section the authors should more discuss the role of SPDEF gene for normal cell development and survival and dual function of SPDEF dependent on molecular subtypes of breast cancer. As the data on the role of SPDEF in breast cancer have been controversial over the years this should be better addressed.
Author Response
Reviewer 2
Study „ Assessing the Tumor Suppressive Impact and Regulatory Mechanisms of SPDEF Expression in Breast Cancer“ by Solanky et al. evaluates the expression of SPDEF transcription factor in breast cancer and its relation with patient survival and expression of genes involved in EMT, DNA replication and repair and immune regulation. The authors associate its reduced expression with poor outcomes, aggressive molecular features, and epigenetic regulation. Epigenetic silencing of SPDEF was in this study associated with overexpression of DNA methyltransferases Co-expression analysis revealed that SPDEF down-regulation coincided with loss of luminal markers and increased EMT, proliferation, DNA repair, and immune pathways.
The study is very good conceptualized and the manuscript is well written.
However, I have few suggestions.
I think that BRCA is not appropriate abbreviation for breast cancer as it is usually related to BRCA1 and BRCA2 genes. Response: Great suggestion. We have replaced all instances of BRCA as abbreviation for breast cancer with "breast cancer."
Also, in Discussion section the authors should more discuss the role of SPDEF gene for normal cell development and survival and dual function of SPDEF dependent on molecular subtypes of breast cancer. As the data on the role of SPDEF in breast cancer have been controversial over the years this should be better addressed. Response: Very good suggestion. We have addressed this in the discussion section by adding extra paragraph.
Round 2
Reviewer 1 Report
Comments and Suggestions for Authors
Please note that in Table 2 there is an incorrect number entry in the "ER Positive" row.
Author Response
Please note that in Table 2 there is an incorrect number entry in the "ER Positive" row
Response: It is fixed now